# Developmental stage shapes the realized energy landscape for a flight specialist

Elham Nourani[1,2]*, Louise Faure[1,2,3]†, Hester Brønnvik[1,2]†, Martina Scacco[1,2]†, Enrico Bassi[4], Wolfgang Fiedler[1,2], Martin U Grüebler[5], Julia S Hatzl[5,6], David Jenny[5], Andrea Roverselli[4], Petra Sumasgutner[7], Matthias Tschumi[5], Martin Wikelski[1,2], Kamran Safi[1,2]

[1]Department of Migration, Max Planck Institute of Animal Behavior, Radolfzell, Germany; [2]Department of Biology, University of Konstanz, Konstanz, Germany; [3]Section Géographie, École normale supérieure de Lyon, Lyon, France; [4]ERSAF-Direzione Parco Nazionale dello Stelvio, Bormio, Italy; [5]Swiss Ornithological Institute, Sempach, Switzerland; [6]Landscape Ecology Institute of Terrestrial Ecosystems , ETH Zürich, Zürich, Switzerland; [7]Konrad Lorenz Research Center (KLF), Core Facility for Behavior and Cognition, Department of Behavioral and Cognitive Biology, University of Vienna, Grünau/Almtal, Austria

*For correspondence:
enourani@ab.mpg.de

†These authors contributed equally to this work

**Abstract** The heterogeneity of the physical environment determines the cost of transport for animals, shaping their energy landscape. Animals respond to this energy landscape by adjusting their distribution and movement to maximize gains and reduce costs. Much of our knowledge about energy landscape dynamics focuses on factors external to the animal, particularly the spatio-temporal variations of the environment. However, an animal's internal state can significantly impact its ability to perceive and utilize available energy, creating a distinction between the 'fundamental' and the 'realized' energy landscapes. Here, we show that the realized energy landscape varies along the ontogenetic axis. Locomotor and cognitive capabilities of individuals change over time, especially during the early life stages. We investigate the development of the realized energy landscape in the Central European Alpine population of the golden eagle *Aquila chrysaetos*, a large predator that requires negotiating the atmospheric environment to achieve energy-efficient soaring flight. We quantified weekly energy landscapes using environmental features for 55 juvenile golden eagles, demonstrating that energetic costs of traversing the landscape decreased with age. Consequently, the potentially flyable area within the Alpine region increased 2170-fold during their first three years of independence. Our work contributes to a predictive understanding of animal movement by presenting ontogeny as a mechanism shaping the realized energy landscape.

## eLife assessment

This **important** study substantially advances our understanding of energy landscapes and their link to animal ontogeny. The evidence supporting the conclusions is **compelling**, with high-throughput telemetry data and advanced track segmentation methods used to develop and map energy land-scapes. The work will be of broad interest to animal ecologists.

## Introduction

The physical characteristics of the environment determine the energetic cost of transport for animals, defining their 'energy landscape' (*Wilson et al., 2012*; *Shepard et al., 2013*). Consequently, temporal dynamics in environmental conditions can affect the inherently spatially explicit energy landscapes

across time. For example, patterns of thawing and freezing of lakes and rivers shape the migratory routes of Caribou, as walking on ice is energetically preferable to swimming (*Leblond et al., 2016*). Variations of the energy landscape along the temporal axis are most notable for animals that move in moving media: water or air. Sharks adjust their core habitat use to make the most of temporary updrafts formed by tidal currents colliding with upward-facing slopes, which help them reduce the energetic costs of remaining buoyant (*Papastamatiou et al., 2021*). Temporal variations in atmospheric conditions shape hourly (*Shepard et al., 2013*) and seasonally (*Nourani et al., 2016*) optimal flight routes for birds.

The spatio-temporal variation in the energy landscape is predictable to a great extent. Animals reduce and thus optimize the cost of transport by tracking these changes as they move through the energy landscapes. As such, the distribution and movement patterns of animals can be largely explained by the energy landscape (*Shepard and Lambertucci, 2013*), making energy landscapes useful for understanding ecological and evolutionary processes. This knowledge is increasingly important for policymaking to reduce human–wildlife conflicts, particularly for development of infrastructures (*Péron et al., 2017*; *Scacco et al., 2023*), and more generally in conservation planning (*Berti et al., 2023*). Based on the timing and location of movement, the energy landscape can be specific yet predictable for animals with similar morphology and movement modes, at the guild (*Shepard et al., 2013*; *Masello et al., 2021*), species (*Scacco et al., 2023*), or sub-species (*Nourani et al., 2020*) levels.

To date, our understanding of animals' interaction with the energy landscape has depended on factors extrinsic to the moving animal. Reconstructing the energy landscape solely based on the characteristics of the environment can be conceptualized as similar to Hutchinson's concept of the fundamental ecological niche (*Hutchinson, 1957*; *Colwell and Rangel, 2009*). Hutchinson defined the fundamental niche as the full range of environmental conditions in which a species can potentially exist and reproduce. Here, we argue that the energy landscape too can, or possibly should, be conceptualized as the fundamental movement niche, defining the full range of physical environmental conditions within which a species can move while adhering to its movement-related energy budget or sustainable cost of transport. However, just as animals do not occupy the entirety of their fundamental Hutchinsonian niche in reality (*Colwell and Rangel, 2009*), for example, due to competition or predation risk, various factors can contribute to an animal not having access to the entirety of its fundamental movement niche. Our current understanding of the energy landscape lacks a differentiation between a fundamental and a realized energy landscape.

An animal's realized energy landscape should depend on the motive to move and its internal state. For example, an animal's decision-making when moving might be the outcome of its strategy to maximize long-term fitness rather than short-term energy savings. *Halsey, 2016* introduced the concept of 'individual energy landscapes', stating that the energy landscape fluctuates, for example, when an animal increases its speed to out-run a predator (*Halsey, 2016*) or when infection or illness reduces an animal's movement capacity (*Binning et al., 2017*; *Risely et al., 2018*). As a consequence, the realized energy landscape varies in time from the perspective of the moving animal, sometimes irrespective of, but in interaction with and as a consequence of, the immediate environment.

We would like to argue that the animals' ability to exploit the energy landscape is transformed across one of the most fundamental progressions in all animals' lives: the ontogenetic axis. In addition to morphological changes, as young animals progress through their developmental stages, their movement proficiency (*Corbeau et al., 2020*) and cognitive capabilities (*Fuster, 2002*) improve and memory manifests (*Ramsaran et al., 2019*). Cognitive capabilities allow the animal to improve the perception of its environment and make optimal decisions when choosing energy-efficient paths (*Harten et al., 2020*; *Abrahms et al., 2021*). Improved motor skills enable the animal to respond more appropriately to the perceived environment to utilize the maximum available energy and/or to avoid areas expensive to traverse (*Scott et al., 2014*; *Sergio et al., 2014*; *Harel et al., 2016*; *Sergio et al., 2022*). Finally, memory allows the animal to recall past combinations of space and environmental conditions that affected its costs of movement (*Abrahms et al., 2019*). Developmental changes and improvements in all these elements should be reflected in the animal's realized energy landscape, where hotspots of energy gain within the landscape expand as young animals gain experience and master their specific movement behaviors.

We investigated the demographic shift in the realized energy landscape by gaining experience as a movement specialist, the golden eagle *Aquila chrysaetos*. As obligate soaring birds, golden eagles utilize ascending air currents, known as uplifts, to gain vertical altitude, which subsequently enables them to glide for horizontal displacement, without relying on energetically expensive wing flapping. Soaring flight is a learned and acquired behavior (*Harel et al., 2016*; *Ruaux et al., 2020*), requiring advanced cognitive skills to locate uplifts as well as fine-tuned locomotor skills for optimal adjustment of the body and wings to extract the most energy from them, for example, by adopting high bank angles in narrow and weak thermals (*Williams et al., 2018a*) and reducing gliding airspeed when the next thermal has not been detected (*Williams et al., 2018b*). This flight mode enables soaring birds to cover long distances with minimal energy investments, almost as low as when resting (*Duriez et al., 2014*). We approach the energy landscape concept with a focus on energy availability. This is commonly done for soaring birds, as uplift and wind support can directly provide energy for flying birds (*Nourani et al., 2020*; *Scacco et al., 2019*).

We analyzed 46,000 hr of flight data collected from bio-logging devices attached to 55 wild-ranging golden eagles in the Central European Alps. These data covered the transience phase of natal dispersal (hereafter post-emigration). In this population, juveniles typically achieve independence by emigrating from the parental territory within 4–10 months after fledging. However, due to the high density of eagles and consequently the scarcity of available territories, the transience phase between emigration and settling by eventually winning over a territory is exceptionally long at well over 4 years. Our hypothesis posited that the realized energy landscape during this transience phase gradually expands as the birds age. More specifically, we expected the habitat to become progressively cheaper to traverse as the birds aged, resulting in an expansion of flyable areas within the Alpine region as a consequence of gaining experience and flight proficiency.

## Results

We explored the movement decisions of 55 juvenile golden eagles in the Central European Alps during their post-emigration commuting flights (i.e., nonstop flight bouts lasting at least 1 hr) from the first week until 3 years after emigration. Following a step-selection approach (*Avgar et al., 2016*), at each movement step, we compared the observed step to 50 alternative steps that were available to each bird in space and time. These steps were compared with respect to the topography of the terrain, specifically Topographic Ruggedness Index (TRI) and distance to ridge lines, both useful proxies for occurrence of uplifts (*Scacco et al., 2019*; *Murgatroyd et al., 2018*). The resulting step-selection model distinguishes between the used and available conditions with a good predictive performance based on the normalized root mean squared error (RMSE = 0.14). Overall, the birds preferred to fly over areas with high potential for uplift formation, characterized by high values of TRI and lower distance to ridge lines (*Figure 1*). The birds also preferred to fly with long step lengths, a trend that increased with age. As the birds grew older, they were less likely to avoid areas with lower TRI and

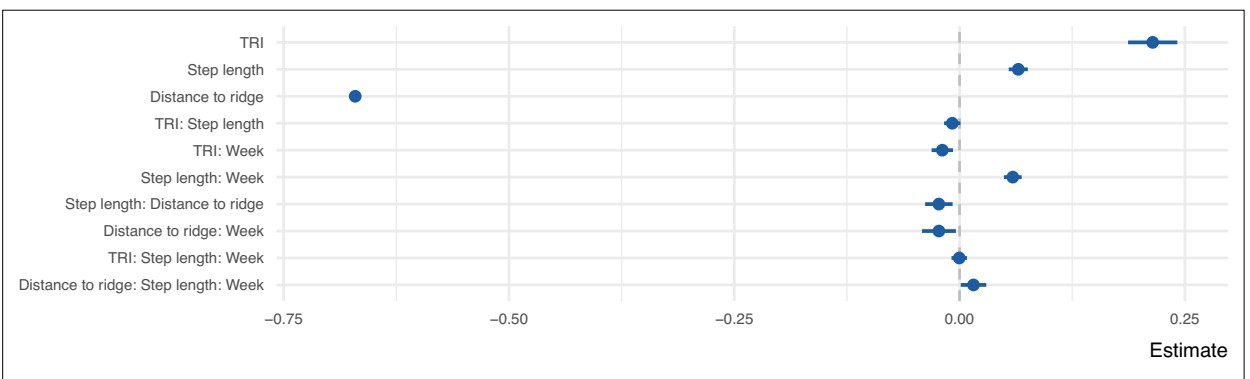

**Figure 1.** Coefficient estimates of the step selection function predicting the probability of use as a function of uplift proxies, week since emigration, and step length. All variables were z-transformed prior to modeling. The error bars show 95% confidence intervals.

The online version of this article includes the following figure supplement(s) for figure 1:

**Figure supplement 1.** Individual-specific slopes for Topographic Ruggedness Index (TRI) and distance to ridge line.

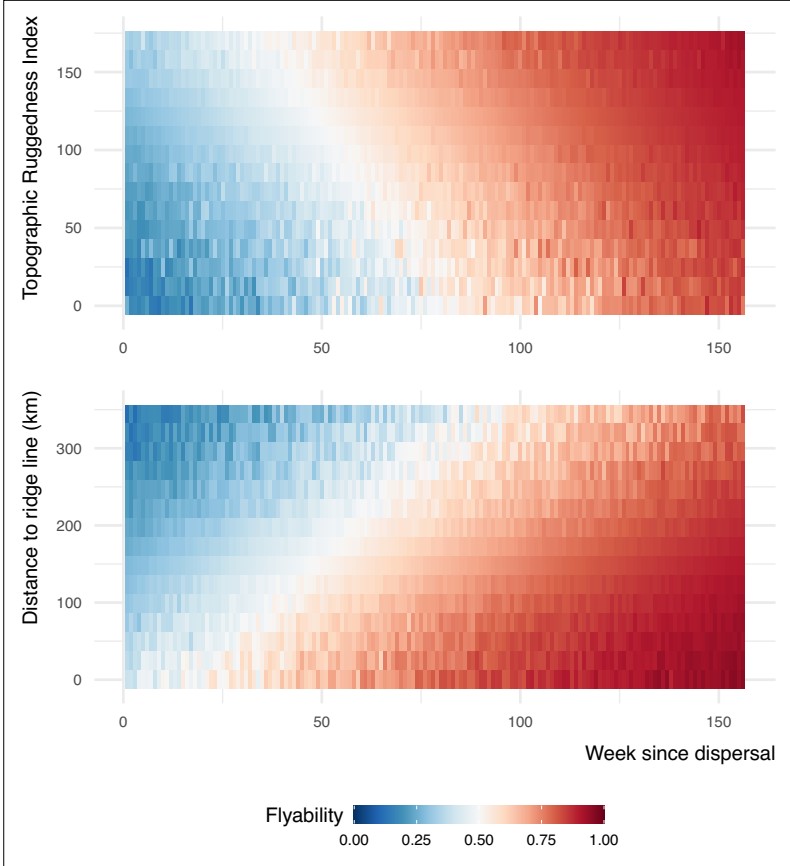

**Figure 2.** Flyability index predicted using the step-selection model for combinations of topography and week since emigration values. The interactions between Topographic Ruggedness Index (TRI) and distance to ridge lines with week are among the set of criteria that young eagles used for selecting where to fly during their commuting flights.

higher distance to ridge lines (*Figure 2*). We found individual variation in the coefficients for TRI and distance to ridge line (*Figure 1—figure supplement 1*).

We used the step-selection model to predict the flyability of the Alpine region for each developmental stage (week since emigration). This flyability index quantified the potential energy that a bird could obtain from its surroundings by considering topography and the bird's movement. This index serves as a measure of the suitability of a location for energy-efficient flight, reflecting the bird's ability to exploit favorable uplift conditions. We constructed energy landscapes for the juvenile golden eagles for each week since independence by predicting the flyability within the Alpine region. Based on these predictions, we determined the flyable area for the juvenile and immature golden eagles during the developmental phase at weekly increments. We defined flyable areas as the cells within a 100 * 100 m grid with a flyability value larger than 0.7. Our analysis shows that hotspots of energy availability in the birds' landscape expanded over time (*Figure 3*, *Video 1*). We found that the flyable area increased 2170-fold from the first week until 3 years after emigration. From an initial 0.038% in the first week after emigration, the flyable area followed a logistic growth curve to plateau at 81% of the entire Alpine region becoming flyable in the third year (*Figure 4*). The tracking data indicated that the golden eagles flew over an area of 92,000 km$^2$ by the third year after emigration (*Figure 4—figure supplement 1*).

## Discussion

We show that the flyable area for juvenile golden eagles increased during the developmental period. This increase, we argue, is most likely linked to the birds' improving ability to negotiate their atmospheric environment better as they aged due to an increasing capacity to perceive and exploit the

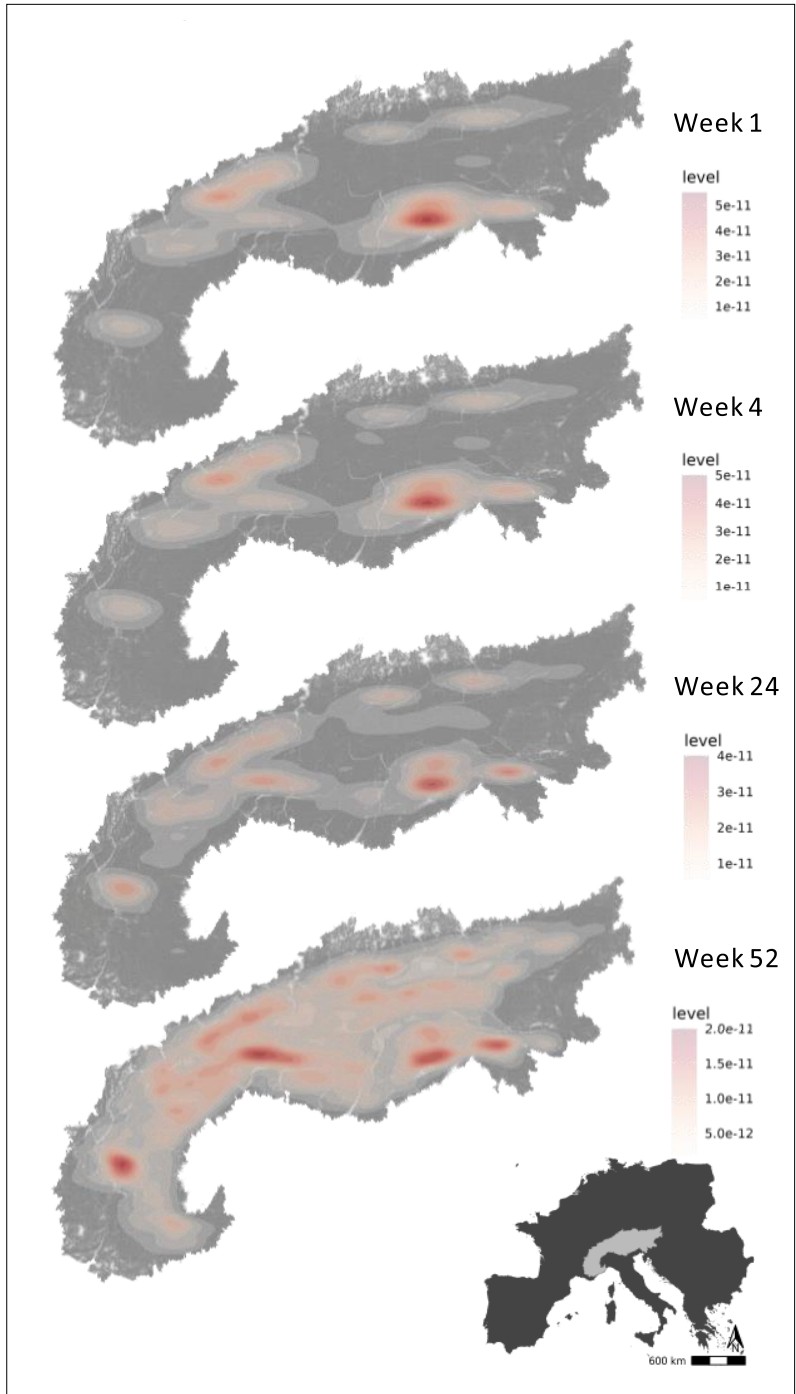

**Figure 3.** Hotspots of energy availability for golden eagles' flight in the Alps. Flyable areas were defined as cells within a 100 * 100 m grid with predicted flyability above 0.7 based on our step-selection model. The maps show the 2D kernel density estimation of flyable areas for golden eagles at different timestamps since dispersal: week 1, week 4 (1 month), week 24 (6 months), and week 52 (1 year). The raw prediction maps for every week since dispersal are shown in *Video 1*.

energy available in the landscape. Because we estimated flyability based on proxies of uplift formation, an important component shaping the energy availability landscape for soaring birds (*Nourani et al., 2020*; *Scacco et al., 2019*), the gradual increase in flyable areas can be interpreted as a change in the realized energy landscape.

Week 1 after emigration

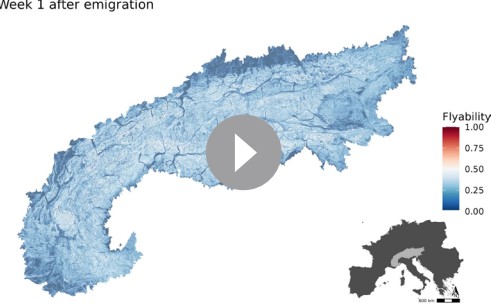

**Video 1.** The realized energy landscape of the aging golden eagles. We used a step-selection approach to determine how juvenile golden eagles responded to topographic conditions during their commuting flights. Topography is a predictor of uplift potential and can be used as a proxy for energy availability for soaring birds. We used a step-selection model to predict flyability across the Alpine region from 1 week to 3 years after emigration. The fundamental energy landscape, defined as the total amount of energy available in the landscape, is constant, but the realized energy landscape, here estimated as flyability, changes. This is because the birds' ability to perceive and exploit the energy within the landscape improves as they age, making the landscape cheaper to traverse. Flyability quantifies the suitability of a location for efficient flight, with higher values indicating areas where the bird is more likely to benefit from favorable uplifts. It represents the realized energy landscape that a bird can exploit based on the given topographic conditions and its own cognitive and locomotor abilities.

https://elifesciences.org/articles/98818/figures#video1

Our findings assert the concept of 'individual energy landscapes'. Initially described by *Halsey, 2016*, this concept suggests that the energy expenditure of an animal when moving is more complex than the cost of transport as determined solely by its morphology and the external environment. The locomotor decisions that an animal makes to maximize its fitness at any given point in time would also affect its energy expenditure. We present evidence that ontogeny also serves as a potential mechanism giving rise to individual energy landscapes not only due to its impact on decision-making, but also due to the influence it exerts on the available options for the animal. During the early stages of their development, animals may not possess the full capacity to perceive their environment, as they will in later stages (*Harten et al., 2020*; *Abrahms et al., 2021*). Additionally, their motor skills are still being honed (*Harel et al., 2016*). Consequently, younger individuals may respond less efficiently to the environmental conditions they must navigate, in contrast to older, more proficient individuals.

By relying on static topographic variables to build the energy landscapes, we ensured that the environment remained constant over time. Juvenile golden eagles complete their morphological development before gaining independence from their parents, with their size and wing morphology remaining stable during the post-emigration phase (*Bortolotti, 1984*; *Katzner et al., 2020*). Consequently, variations in flyability

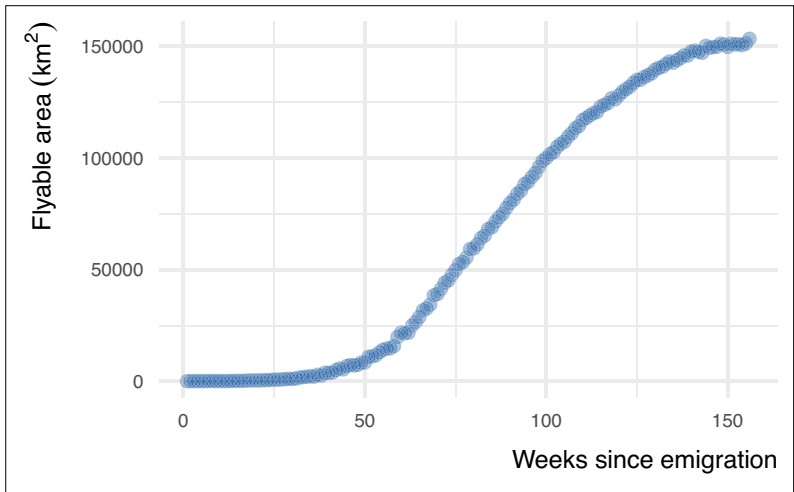

**Figure 4.** The flyable area for juvenile golden eagles in the Alpine region from the first week until 3 years after emigration. Flyable area was defined as the total number of cells within a 100 * 100 m grid with predicted flyability larger than 0.7 based on the step-selection model. The positive trend shows that juvenile golden eagles can fly over a larger portion of the Alpine region as they age.

The online version of this article includes the following figure supplement(s) for figure 4:

**Figure supplement 1.** Cumulative area used by juvenile golden eagles during each week after emigration.

of the landscape for these birds predominantly reflect their improved mastery of soaring flight, rather than changes in their morphology. Our findings also reveal that as the eagles aged, they adopted longer step lengths, which could indicate an increasing ability to sustain longer uninterrupted flight bouts.

Furthermore, our results provide insight into the development of different flight modes. These birds can switch between orographic and thermal soaring (*Katzner et al., 2015*). Orographic soaring refers to the utilization of upward air currents generated when wind encounters and flows over elevated topographic features such as hills, mountains, or cliffs. Locating orographic uplifts requires less experience, as they form predictably at mountain ridges (*Bohrer et al., 2012*; *Duerr et al., 2012*), compared to locating and using more transient thermals which form less predictably over flat surfaces (*Scacco et al., 2019*). Our findings suggest that with increasing age, golden eagles place decreasing importance on ruggedness and proximity to ridge lines when deciding where to fly, indicating a learning progression from mastering orographic uplifts before tackling the more challenging thermal soaring.

The energy landscapes maps showed the amount of energy potentially available to juvenile golden eagles during progressing weeks since emigration (*Video 1*). These maps are based on a step-selection model built on the birds' empirical use of the landscape, the energy availability thus represents the energy that would be theoretically perceivable and exploitable by the birds in a specific developmental phase or age. The detailed perspective on an energy landscape which transforms over time with the accumulation of experience highlights the process of changes in the realized movement niche that would be available to the animals at a certain age. These changes in the ability to extract energy from the atmospheric column will induce age-specific consequences on movement capacity and the cost of transport.

We argue that the realized energy landscape does not represent the realized ecological niche nor the distribution range of animals, but rather the subset of the fundamental energy landscape that is available to the individual based on its ability to perceive and exploit the energy in the landscape. In our study, the realized movement patterns of the golden eagles within the Alpine region covered a small portion of the area predicted as flyable. This is because the birds' movement (and thus their realized individual and age-specific movement niches) will be defined not only by the energy landscape, but also by the interplay of philopatry, developmental requirements, the social landscape, and anthropogenic features. As golden eagles age, other crucial behaviors such as transitioning from predominantly scavenging to active hunting are also happening (*Nygård et al., 2016*). A transition that in itself also might depend on advanced flying skills. Thus, the choice of where to fly will not solely follow an energy-minimizing strategy while moving, but naturally also increasingly accommodate resource use opportunities. Additionally, older transient golden eagles are more susceptible to potentially deadly conflicts with territorial adult birds, influencing their movement decisions as they avoid established territories (*Poessel et al., 2016*) affecting their landscape of fear and thus their decision landscape overall (*Laundré et al., 2001*; *Gallagher et al., 2017*; *Williams and Safi, 2021*). The distribution of anthropogenic features could also impact the eagles' decision-making on where to fly (*Tack et al., 2020*). The strong individual variation that we observed in the birds' response to distance to ridge lines (*Figure 1—figure supplement 1*) could indicate variable landscape utilization strategies among the juvenile golden eagles in response to these factors and their interaction.

## Conclusion

Spatial maps serve as valuable tools in informing conservation and management strategies by showing the general distribution and movement patterns of animals. These tools are crucial for understanding how animals interact with their environment, including human-made structures. Within this context, energy landscapes play an important role in identifying potential areas of conflict between animals and anthropogenic infrastructures such as wind farms. The predictability of environmental factors that shape the energy landscape has facilitated the development of these conservation tools, which have been extrapolated to animals belonging to the same ecological guild traversing similar environments. However, with recent development of concepts of species-specific (*Scacco et al., 2023*) and individual-specific energy landscapes (*Halsey, 2016*), the realistic prediction of animals' energy landscapes appears a multifaceted challenge. By identifying the distinct dimensions along which individual energy landscapes can vary, particularly along the demographic axis, we enhance our capacity to understand and predict these landscapes. Our research contributes to this broader understanding

by revealing the stark influence of ontogeny in defining the aging of the realized energy landscape. By taking inspiration from the ecological niche definitions to conceptualize the energy landscape, we hope to improve further development of this concept, aiding the interpretation of variations in the energy landscape across many axes, both external and internal to the moving animal.

## Materials and methods

### Bio-logging dataset

We used Global Positioning System tracking data from a long-term study of juvenile golden eagles in the resident population of the Central European Alps. We used data collected for a total of 55 juveniles tracked over 2017–2023 (*Supplementary file 1*) with a regular sampling schedule of one GPS fix every 20 min during day time (see *Zimmermann, 2021* for details on fieldwork procedure and data sampling schedules). Nestlings were fitted with solar-powered loggers manufactured by e-obs GmbH (Munich, Germany) using a leg-loop harness. The weight of the harness and logger (in total maximum 60 g) was below the recommended maximum 3% of the birds' body mass. We subset the data to correspond to the transient phase of natal dispersal up to 3 years after emigration. We identified emigration timing using recursions to the natal territory (*Bracis et al., 2018*). If an animal spent more than 14 consecutive days without recursions to a 7 km radius around its nest, we classified it as dispersed. These thresholds were chosen based on expert knowledge of golden eagle behavior in the region. For two animals that had exceptionally large natal territories to start with, we increased the radii to 30 km to compensate for this potential bias.

### Track segmentation

We focused on commuting flights for investigating the energy landscape for the golden eagles. To extract commuting flights, we segmented the tracking data based on ground speed and flight altitude using Expectation-Maximization binary Clustering (*Garriga et al., 2016*) applied at the population level (i.e., to all individuals). To calculate flight altitude above ground for each tracking point, we first calculated altitude above mean sea level as the difference between altitude above ellipsoid recorded by the bio-logging devices and the Global Geoid Model EGM96 (WGS 84). We then subtracted the ground elevation (Copernicus Digital Elevation Model; 25 m resolution; https://land.coper-nicus.eu/) from altitude above mean sea level as an estimate of flight altitude. We labeled segments with high flight altitude and high speed as commuting flights and used these for further analysis. We sub-sampled the tracking data to hourly intervals to focus on commuting flights that were longer than 1 hr.

### Topographic variables

We used two topographic variables as proxies for uplift availability: TRI and distance to ridge lines. The former describes the topographic heterogeneity of the terrain by calculating the mean of the absolute elevation difference between a cell and its adjacent cells (*Riley et al., 1999*) and the latter represents a proxy for orographic updrafts generated by deflection of the air current by a physical barrier (*Murgatroyd et al., 2018*; *Bohrer et al., 2012*). To extract the ridge lines and TRI values, we used the Copernicus Digital Elevation Model. We identified the ridge lines based on the Weisss landforms classification using TPI, a scale-dependent index which compares the elevation of each cell to the neighboring cells within a given radius (*Weiss, 2001*). As the landform categories depend on the size of the radius, several TPI classifications at different scales were obtained using different radii (*Das et al., 2015*; *Ilia et al., 2016*). We selected a classification using a 10 m inner radius and 200 m outer radius on the basis of the narrow size of the identified ridge lines. Then, we calculated the distance from each pixel to the ridge line pixel using the function proximity raster from SAGA (*Conrad et al., 2015*) to obtain a distance map to the ridge line. For both TRI and distance to ridge lines, we averaged neighboring cell values to achieve a 100 m cell size.

### Modeling the energy landscape

We used a step-selection approach (*Avgar et al., 2016*) to summarize the environmental conditions in which the birds chose to commute. Each pair of consecutive points in the tracking dataset was considered as one step. For each step, we generated 50 random steps that originated at the observed steps' start point, but had alternative end points to represent alternative movement decisions that

the bird could have taken. The location of the alternative end points was selected by drawing from a gamma distribution fitted to the step lengths and a von Mises distribution fitted to the turning angles of all the empirical tracking data. The result was a stratified dataset with one used and 50 alternative points in each stratum.

All used and available points were associated with TRI and distance to ridge line values. We found no autocorrelation ($r < 0.5$) among our predictor variables TRI, distance to ridge line, step length, and week since emigration (as a continuous variable). The predictor variables were z-transformed prior to modeling to ensure comparability of the coefficient estimates. We used a conditional logistic regression to build a step-selection function to model and predict the energy landscape. We included step length as one of the predictors in our model to take the movement characteristics of the growing juveniles into account (*Avgar et al., 2016*). To track the changes in the importance of the topographic variables over time, we included three-way interaction terms for the TRI and distance to ridge line, with step length and week since emigration. To account for individual variation, we included the random effect of individuals on the slopes of TRI and distance to ridge line. The model was fit using the Template Model Builder (*Brooks et al., 2017*) in R (*R Development Core Team, 2022*), following the procedure suggested by *Muff et al., 2020*.

To test the predictive performance of the model, we computed the normalized RMSE (*Lüdecke et al., 2021*). This metric can be interpreted as the standard deviation of the unexplained variance, with lower values indicating a smaller difference between observed data and the model's predicted values.

To better interpret the interaction terms between the topographic variables and week since emigration, we made predictions using the model for different combinations of TRI and week as well as distance to ridge line and week. In the new datasets that we generated to make these predictions, we set the variables that were not represented in the interaction term to their average values and set the grouping factor that represented the stratum as missing data. We made the predictions on the scale of the link function and converted them to values between 0 and 1 using the inverse logit function (*Bolker et al., 2022*). These predicted values estimated the probability of use of an area based on the model. We interpreted these predicted values as the flyability index, representing the potential energy available in the landscape to support flight, based on the uplift proxies (TRI and distance to ridge line) and the movement capacity (step length) of the birds included in the model.

## Energy landscape maps

To investigate the changes in the energy landscape across the ontogenetic axis, we used our model to create prediction maps for weekly increments during the first 3 years after emigration. We made our prediction maps for the entire extent of the Alpine region ('Perimeter of the Alpine Convention' layer available via Permanent Secretariat of the Alpine Convention). For each week since emigration, we created a new dataset to make predictions for each 100 m × 100 m cell. We assigned the mean step length value for each given week in the new datasets.

To explore the changes in flyable areas over time, we defined landscape flyability as flyability values above 0.7. We then estimated the total flyable areas for each week since emigration. The Alpine region covers 190,544.7 km² (based on the Alpine perimeter layer mentioned above). As a final step, we compared the eagles' flyable area in weeks 1 and 156 with the total area of the Alpine region.

## Acknowledgements

EN was supported by the PRIME program of the German Academic Exchange Service (DAAD) with funds from the German Federal Ministry of Education and Research (BMBF). LF was partially supported by the Deutsche Forschungsgemeinschaft (DFG, German Research Foundation) under Germany's Excellence Strategy EXC 2117 422037984 (grant M21-20 to EN). HB was partially supported by the DFG (Emmy Noether Fellowship 463925853 to Andrea Flack) and the International Max Planck Research School for Quantitative Behaviour, Ecology and Evolution (IMPRS-QBEE). We thank Michael Chimento for scientific and technical discussions. We are grateful for the crucial assistance of the mountain rescue teams of the Guardia di Finanza as well as the forestry and game wardens of South Tyrol/Alto Adige and the national park team in Gesäuse, as well as the Stelvio national park Italy and the Fish and Game Department of the Canton of Grisons (AJF) and many gamekeepers. We also thank the accompanying veterinarians Drs. Michel Mottini and Gilberto Volcan for their time and expertise in

the field. The following people assisted with fieldwork in Slovenia: Teja Curk, Toma Miheli, Bor Miheli, Gaber Miheli, Ruj Miheli and Miha nidari; in Italy: Adriano Greco, Klaus Bliem, Martin Trafoier, Claudio Angeli, Italo Armanasco, Anna Bonettini, Cristian Capitani, Paola Chiudinelli, Alessandro Mercogliano, Alberto Pastorino, Laura Tomasi, Antonella Cordedda and i carabinieri forestali, Jagdverband Südtirol; in Austria: Shane Sumasgutner; in Germany: Ulrich Brendel, Jochen Grab, Wolfgang Palzer; in Switzerland: Claudio Schorta, Claudia Gerber, Svea Zimmermann, Sam Cruickshank, Hannes Jenny, and the climbers Romano Salis, Geni Ballat, Stefan Rauch and Carlo Micheli. We are grateful to anonymous reviewers for their insightful comments on previous versions of this work, which helped us refine our story. Open access funding was provided by the Max Planck Society.

## Additional information

### Funding

| Funder | Grant reference number | Author |
|---|---|---|
| German Academic Exchange Service | | Elham Nourani |
| Deutsche Forschungsgemeinschaft | EXC 2117 - 422037984 | Elham Nourani Louise Faure |
| Deutsche Forschungsgemeinschaft | Emmy Noether Fellowship 463925853 | Hester Brønnvik |
| International Max Planck Research School for Quantitative Behaviour, Ecology and Evolution | | Hester Brønnvik |

The funders had no role in study design, data collection and interpretation, or the decision to submit the work for publication. Open access funding provided by Max Planck Society.

### Author contributions

Elham Nourani, Conceptualization, Formal analysis, Funding acquisition, Validation, Investigation, Visualization, Methodology, Writing – original draft, Project administration, Writing – review and editing; Louise Faure, Formal analysis, Methodology, Writing – original draft, Writing – review and editing; Hester Brønnvik, Martina Scacco, Formal analysis, Methodology, Writing – review and editing; Enrico Bassi, Martin U Grüebler, Julia S Hatzl, David Jenny, Andrea Roverselli, Petra Sumasgutner, Matthias Tschumi, Data curation, Writing – review and editing; Wolfgang Fiedler, Data curation, Project administration, Writing – review and editing; Martin Wikelski, Data curation, Funding acquisition, Writing – review and editing; Kamran Safi, Conceptualization, Data curation, Validation, Writing – original draft, Project administration, Writing – review and editing

### Author ORCIDs

Elham Nourani ⓘD https://orcid.org/0000-0003-4420-3902
Louise Faure ⓘD https://orcid.org/0009-0007-4504-3310
Hester Brønnvik ⓘD https://orcid.org/0000-0002-3061-0630
Martin U Grüebler ⓘD https://orcid.org/0000-0003-4468-8292
Petra Sumasgutner ⓘD https://orcid.org/0000-0001-7042-3461
Matthias Tschumi ⓘD https://orcid.org/0000-0002-7991-7780
Kamran Safi ⓘD https://orcid.org/0000-0002-8418-6759

### Ethics

The handling and ringing of golden eagle nestlings in Switzerland was carried out under the authorization of the Office for Food Safety and Animal Health (ALT) of the canton Grisons (licence No. GR2017_06, GR 2018_05E, GR 2019_03E). In Italy, the permissions for handling, tagging and marking were obtained from autonomous region of South Tyrol (Dekret 12257/2018 and Dekret 8788/2020), as well as from the region of Lombardia for ringing and tagging through in Lombardia and South Tyrol by ISPRA (Istituto Superiore per la Protezione e la Ricerca Ambientale) with the Richiesta di

autorizzazione alla cattura di fauna selvatica per scopi scientifici (l.r. 26/93). In Austria, all procedures for handling, tagging and marking were approved by the Ethics Committee of the University of Vienna (No. 2020-008) and permitted by the Federal Ministry for Education, Science and Research (No. 2020-0.547.571), Styria (BHLI-165942/2021-2) and Upper Austria (LFW-2021-263262/7-Sr). Finally, in Germany birds handled, tagged and ringed were done so under the permission issued by the government of Oberbayern (2532.Vet_02-16-88 and 2532.Vet_02-20-86). All procedures followed the ASAB/ABS guidelines for the ethical treatment of animals in behavioral research and teaching and all applicable international, national, and/or institutional guidelines for the care and use of animals were followed. The handling of birds was performed with maximum care and minimal disturbance to nests and the landscape. Ethical approval for involving animals in this study was received through the application procedure for ringing permits and the scientific commission of the Swiss Ornithological Institute and the national authorities as well as the guidelines imposed through the European Commission on the basis of the guidelines of the FELASA (the Federation or the European Laboratory Animals Association).

Reviewer #1 (Public review): https://doi.org/10.7554/eLife.98818.3.sa1
Reviewer #2 (Public review): https://doi.org/10.7554/eLife.98818.3.sa2
Author response https://doi.org/10.7554/eLife.98818.3.sa3

## Additional files

### Supplementary files
• Supplementary file 1. Details of bio-logging data included in the study. The country and year of tagging for each individual, and the tracking duration (in terms of weeks since dispersal) that each individual contributed to the analysis, are reported. All individuals carried Bird Solar Tags manufactured by e-obs GmbH, Germany (either 45 gr or 25 gr devices). Individual local identifiers match those included in Movebank studies 'LifeTrack Golden Eagle Alps' and 'LifeTrack Golden Eagle Alps Public'.

• MDAR checklist

### Data availability
All data needed to replicate the findings of this article can be accessed via an Edmond repository 'Energy landscape Ontogeny' (https://doi.org/10.17617/3.FM4EJC). The code for data preparation, analysis, and plotting can be found in the GithHub repository 'golden_eagle_energy_landscape' (https://github.com/mahle68/golden_eagle_energy_landscape/, copy archived at *Nourani, 2024*).

The following dataset was generated:

| Author(s) | Year | Dataset title | Dataset URL | Database and Identifier |
|---|---|---|---|---|
| Nourani E | 2023 | Energy landscape ontogeny | https://doi.org/10.17617/3.FM4EJC | Edmond, 10.17617/3.FM4EJC |

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
