## [Editor Report · eLife assessment]

This **important** study substantially advances our understanding of energy landscapes and their link to animal ontogeny. The evidence supporting the conclusions is **compelling**, with high-throughput telemetry data and advanced track segmentation methods used to develop and map energy landscapes. The work will be of broad interest to animal ecologists.

---

## [Referee Report · Reviewer #1 (Public review)]

Summary:

The authors propose that the energy landscape of animals can be thought of in the same way as the fundamental versus realized niche concept in ecology. Namely, animals will use a subset of the fundamental energy landscape due to a variety of factors. The authors then show that the realized energy landscape of eagles increases with age as the animals are better able to use the energy landscape.

Strengths:

This is a very interesting idea and that adds significantly to the energy landscape framework. They provide convincing evidence that the available regions used by birds increase with size.

Review of revised version:

The authors have addressed all my comments and concerns. This is a really nice and important manuscript. I have one minor suggestion: Line 74-85: when discussing the effect of ontogeny, the authors give examples of how these may change due to improved cognition and memory. I would recommend they also give examples of how these may change with morphology (e.g. change in wing or fin relative area, buoyancy in sharks etc) should also be included. Most growth in fish for example is allometric so the relative measures of area of fins to body size should also change.

This is of course up to the authors but it would highlight how their study is applicable to many other systems beyond just birds (even though morphology is of little importance for their eagles).

---

## [Referee Report · Reviewer #2 (Public review)]

Summary:

With this work, the authors tried to expand and integrate the concept of realized niche in the context of movement ecology by using fine-scale GPS data of 55 juvenile Golden eagles in the Alps. Authors found that ontogenic changes influence the percentage of area flyable to the eagles as individuals exploit better geographic uplifts that allow them to reduce the cost of transport.

Strengths:

Authors made insightful work linking changes in ontogeny and energy landscapes in large soaring birds that may not only advance the understanding of how changes in the life cycle affect the exploitability of aerial space but also offer valuable tools for the management and conservation of large soaring species in the changing world.

Weaknesses:

Future research may test the applicability of the present work by including more individuals and/or other species from other study areas.

---

## [Author Response]

The following is the authors’ response to the original reviews.

**Public Reviews:**

**Reviewer 1 (Public Review):**
Summary:The authors propose that the energy landscape of animals can be thought of in the same way as the fundamental versus realized niche concept in ecology. Namely, animals will use a subset of the fundamental energy landscape due to a variety of factors. The authors then show that the realized energy landscape of eagles increases with age as the animals are better able to use the energy landscape. Strengths:This is a very interesting idea and that adds significantly to the energy landscape framework. They provide convincing evidence that the available regions used by birds increase with size.Weaknesses:Some of the measures used in the manuscript are difficult to follow and there is no mention of the morphometrics of birds or how these change with age (other than that they don’t change which seems odd as surely they grow). Also, there may need to be more discussion of other ontogenetic changes such as foraging strategies, home range size etc.

We thank reviewer 1 for their interest in our study and for their constructive recommendations. We have included further discussions of these points in the manuscript and outline these changes in our responses to the detailed recommendations below.

**Reviewer 2 (Public Review):**
Summary:With this work, the authors tried to expand and integrate the concept of realized niche in the context of movement ecology by using fine-scale GPS data of 55 juvenile Golden eagles in the Alps. Authors found that ontogenic changes influence the percentage of area flyable to the eagles as individuals exploit better geographic uplifts that allow them to reduce the cost of transport.Strengths:Authors made insightful work linking changes in ontogeny and energy landscapes in large soaring birds. It may not only advance the understanding of how changes in the life cycle affect the exploitability of aerial space but also offer valuable tools for the management and conservation of large soaring species in the changing world.Weaknesses:Future research may test the applicability of the present work by including more individuals and/or other species from other study areas.

We are thankful to reviewer 2 for their encouragement and positive assessment of our work. We have addressed their specific recommendations below.

**Recommendations for the authors:**

**Reviewer 1 (Recommendations For The Authors):**
I found this to be a very interesting paper which adds some great concepts and ideas to the energy landscape framework. The paper is also concise and well-written. While I am enthusiastic about the paper there are areas that need clarifying or need to be made clearer. Specific comments below:Line 64: I disagree that competition is the fundamental driver of the realized niche. In some cases, it may be but in others, predation may be far more important (as an example).

We agree with this point and have now clarified that competition is an example of a driver of the realized niche. We have also included predation as another example:

"However, just as animals do not occupy the entirety of their fundamental Hutchinsonian niche in reality [1], for example due to competition or predation risk, various factors can contribute to an animal not having access to the entirety of its fundamental movement niche."

Intro: I think the authors should emphasize that morphological changes with ontogeny will change the energy landscape for many animals. It may not be the case specifically with eagles but that won’t be true for other animals. For example, in many sharks, buoyancy increases with age.

We agree and have now clarified that the developmental processes that we are interested in happen in addition to morphological changes:

"In addition to morphological changes, as young animals progress through their developmental stages, their movement proficiency [2] and cognitive capabilities [3] improve and memory manifests [4]."

Line 91-93: The idea that birds fine-tune motor performance to take advantage of updrafts is a very important one to the manuscript and should be discussed in a bit more detail. How? At the moment there is a single sentence and it doesn’t even have a citation yet this is the main crux of the changes in realized energy landscape with age. This point should be emphasized because, by the end of the introduction, it is not clear to me why the landscape should be cheaper as the birds age?

Thank you for pointing out this missing information. We have now added examples to clarify how soaring birds fine-tune their motor performance when soaring. These include for example adopting high bank angles in narrow and weak thermals [5] and reducing gliding airspeed when the next thermal has not been detected [6]:

"Soaring flight is a learned and acquired behavior [7, 8], requiring advanced cognitive skills to locate uplifts as well as fine-tuned locomotor skills for optimal adjustment of the body and wings to extract the most energy from them, for example by adopting high bank angles in narrow and weak thermals [5] and reducing gliding airspeed when the next thermal has not been detected [6]."

Results:Line 106: explain the basics of the life history of the birds in the introduction. I have no idea what emigration refers to or the life history of these animals.

Thank you for pointing out the missing background information. We have now added this

information to the introduction:

"We analyzed 46,000 hours of flight data collected from bio-logging devices attached to 55 wild-ranging golden eagles in the Central European Alps. These data covered the transience phase of natal dispersal (hereafter post-emigration). In this population, juveniles typically achieve independence by emigrating from the parental territory within 4-10 months after fledging. However, due to the high density of eagles and consequently the scarcity of available territories, the transience phase between emigration and settling by eventually winning over a territory is exceptionally long at well over 4 years. Our hypothesis posited that the realized energy landscape during this transience phase gradually expands as the birds age."

What I still am having a hard time understanding is the flyability index. Is this just a measure of the area animals actively select and then the assumption that it’s a good region to fly within?

We have modified our description of the flyability index for more clarity. In short, we built a step-selection model and made predictions using this model. The predictions estimate the probability of use of an area based on the predictors of the model. For the purpose of our study and what our predictors were (proxies for uplift + movement capacity), we interpreted the predicted values as the "flyability index". We have now clarified this in the methods section:

"We made the predictions on the scale of the link function and converted them to values between 0 and 1 using the inverse logit function [9]. These predicted values estimated the probability of use of an area for flying based on the model. We interpreted these predicted values as the flyability index, representing the potential energy available in the landscape to support flight, based on the uplift proxies (TRI and distance to ridge line) and the movement capacity (step length) of the birds included in the model."

It might also be useful to simply show the changes in the area the animals use with age as well (i.e. a simple utilization distribution). This should increase in age for many animals but would also be a reflection of the resources animals need to acquire as they get older.

We have now added the figure S2 to the supplementary material. This plot was created by calculating the cumulative area used by the birds in each week after emigration. This was done by extracting the commuting flights for each week, converting these to line objects, overlapping the lines with a raster of 100*100 m cell size, counting the number of overlapping cells and calculating the area that they covered. We did not calculate UDs or MCPs because the eagles seem to be responding to linear features of the landscape, e.g. preferring ridgelines and avoiding valleys. Using polygons to estimate used areas would have made it difficult to ensure that decision-making with regards to these linear features was captured.

In a follow-up project, a PhD student in the golden eagle consortium is exploring the individuals’ space use after emigration considering different environmental and social factors. The outcome of that study will further complete our understanding of the post-emigration behavior of juvenile golden eagles in the Alps.

How much do the birds change in size over the ontogeny measured? This is never discussed.

Thank you for bringing up this question. The morphometrics of juvenile golden eagles are not significantly different from the adults, except in the size of culmen and claws [10]. Body mass changes after fledging, because of the development of the pectoral muscles as the birds start flying. Golden eagles typically achieve adult-like size and mass within their natal territory before emigration, at which time we started quantifying the changes in energy landscape. Given our focus on post-emigration flight behavior, we do not expect any significant changes in size and body mass during our study period. We now cover this in the discussion:

"Juvenile golden eagles complete their morphological development before gaining independence from their parents, with their size and wing morphology remaining stable during the post-emigration phase [10, 11]. Consequently, variations in flyability of the landscape for these birds predominantly reflect their improved mastery of soaring flight, rather than changes in their morphology."

Discussion:Line 154: Could the increase in step length also be due to changes in search strategies with age? e.g. from more Brownian motion when scavenging to Levy search patterns when actively hunting?

This is a very good point and we tried to look for evidence of this transition in the tracking data. We explored the first passage time for two individuals with a radius of 50 km to see if there is a clear transition from a Brownian to a Levy motion. The patterns that emerge are inconclusive and seem to point to seasonality rather than a clear transition in foraging strategy (Author response image 1). We have modified our statement in the discussion about the change in preference of step lengths indicating improve flight ability, to clarify that it is speculative:

**Author response image 1. sa3fig1:** First passage times using a 50 km radius for two randomly selected individuals.

"Our findings also reveal that as the eagles aged, they adopted longer step lengths, which could indicate an increasing ability to sustain longer uninterrupted flight bouts."

Methods:Line 229: What is the cutoff for high altitude or high speed?

We used the Expectation-maximization binary clustering (EMbC) method to identify commuting flights. The EmbC method does not use hard cutoffs to cluster the data. Each data point was assigned to the distribution to which it most likely belonged based on the final probabilities after multiple iterations of the algorithm. Author response image 2 shows the distribution of points that were either used or not used based on the EmbC classification.

**Author response image 2. sa3fig2:** Golden eagle tracking points were either retained (used) or discarded (not used) for further data analysis based on the EmbC algorithm. The point were clustered based on ground speed and height above ground.

Figure 1: The figure captions should stand on their own but in this case there is no information as to what the tests are actually showing.

We have now updated the caption to provide information about the model:

"Coefficient estimates of the step selection function predicting probability of use as a function of uplift proxies, week since emigration, and step length. All variables were z-transformed prior to modeling.

The error bars show 95% confidence intervals."

**Reviewer 2 (Recommendations For The Authors):**
First, I want to congratulate you on this fantastic work. I enjoyed reading it. The manuscript is clear and well-written, and the findings are sound and relevant to the field of movement ecology. Also, the figures are neatly presented and easy to follow.I particularly liked expanding the old concept of fundamental vs realized niche into a movement ecology context. I believe that adds a fresh view into these widely accepted ecological assumptions on species niche, which may help other researchers build upon them to better understand movement "realms" on highly mobile animals in a rapidly changing world.I made some minor comments to the manuscript since it was hard to find important weaknesses in it, given the quality of your work. However, there was a point in the discussion that I feel deserves your attention (or rather a reflection) on how major biological events such as moulting could also influence birds to master the flying and exploitation of the energy landscape. You may find my suggestion quite subjective, but I think it may help expand your idea for future works and, what is more, link concepts such as energy landscapes, ontogeny, and important life cycle events such as moulting in large soaring birds. I consider this relevant from a mechanistic perspective to understand better how individuals negotiate all three concepts to thrive and persist in changing environments and to maximise theirfitness.Once again, congratulations on this excellent piece of research.

We thank the reviewer for their enthusiasm about our work and for bringing up important points about the biology of the species. Our detailed response are below.

MINOR COMMENTS:(Note: Line numbers refer to those in the PDF version provided by the journal).Line 110: Distinguished (?)

corrected

Line 131: Overall, I agree with the authors’ discussion and very much liked how they addressed crucial points. However, I have a point about some missing non-discussed aspects of bird ecology that had not been mentioned.The authors argue that morphological traits are less important in explaining birds’ mastery of flight (thus exploiting all available options in the landscape). However, I think the authors are missing some fundamental aspects of bird biology that are known to affect birds’ flying skills, such as moult.The moulting process affects species’ flying capacity. Although previous works have not assessed moults’ impact on movement capacity, I think it is worth including the influence of flyability on this ecologically relevant process.For instance, golden eagles change their juvenile plumage to intermediate, sub-adult plumage in two or three moult cycles. During this process, the moulting process is incomplete and affects the birds’ aerodynamics, flying capacity, and performance (see Tomotani et al. 2018; Hedenström 2023). Thus, one could expect this process to be somewhat indirectly linked to the extent to which birds can exploit available resources.Hedenström, A. (2023). Effects of wing damage and moult gaps on vertebrate flight performance.Journal of Experimental Biology, 226(9), jeb227355. Tomotani, B. M., Muijres, F. T., Koelman, J., Casagrande, S., & Visser, M. E. (2018). Simulated moult reduces flight performance, but overlap with breeding does not affect breeding success in a longdistance migrant. Functional Ecology, 32(2), 389-401.

We thank the reviewer for bringing up this relevant topic. We explored the literature listed by the reviewer and also other sources. We came to the conclusion that moulting does not impact our findings. In our study, we included data for eagles that had emigrated from the natal territories, with their fully grown feathers in juvenile plumage. The moulting schedule in juvenile birds is similar to that of adults: the timing, intensity, and sequence of feathers being replaced is consistent every year (Author response image 3). For these reasons, we do not believe that moulting stage noticeably impacts flight performance at the scale of our study (hourly flights). Fine details of soaring flight performance (aerodynamics within and between thermals) could differs during moulting of different primary and secondary feathers, but this is something that would occur every time the eagle replaces these feather and we do not expect it to be any different for juveniles. Such fine scale investigations are outside the scope of this study.

**Author response image 3. sa3fig3:** Moulting schedule of golden eagles [12].

Lines 181-182: I don’t think trophic transitions rely only on individual flying skill changes. Furthermore, despite its predominant role, scavenging does not mean it is the primary source of food acquisition in golden eagles. This also depends on prey availability, and scavenging is an auxiliary font of easy-to-catch food.Scavenging implies detecting carcasses. Should this carcass appearance occur in highly rugged areas, the likelihood of detection also reduces notably. This is not to say that there are not more specialized carrion consumers, such as vultures, that may outcompete eagles in searching for such resources moreefficiently.In summary, I don‘t think such transition relies only on flying skills but on other non-discussed factors such as knowledge accumulation of the area or even the presence of conspecifics.Line 183: This is precisely what I meant with my earlier comment.

Thank you for the discussion on the interaction between flight development and foraging strategy. We explored the transition from scavenging to hunting above as a response to Reviewer 1, but did not find a clear transition. This is in line with your comment that the birds probably use both scavenging and hunting methods opportunistically.

Lines 193-195: I will locate this sentence somewhere in this paragraph. As it is now, it seems a bit out of context. It could be a better fit at the end of the first point in line 203.

Thank you for pointing out the issue with the flow. We have now added a transitional sentence before this one to improve the paragraph. The beginning of the conclusion now reads as follows, with the new sentence shown in boldface.

"Spatial maps serve as valuable tools in informing conservation and management strategies by showing the general distribution and movement patterns of animals. These tools are crucial for understanding how animals interact with their environment, including human-made structures. Within this context, energy landscapes play an important role in identifying potential areas of conflict between animals and anthropogenic infrastructures such as wind farms. The predictability of environmental factors that shape the energy landscape has facilitated the development of these conservation tools, which have been extrapolated to animals belonging to the same ecological guild traversing similar environments."

References

(1) Colwell, R. K. & Rangel, T. F. Hutchinson’s duality: The once and future niche. *Proceedings of the National Academy of Sciences* 106, 19651–19658. doi:10.1073/pnas.0901650106 (2009).

(2) Corbeau, A., Prudor, A., Kato, A. & Weimerskirch, H. Development of flight and foraging behaviour in a juvenile seabird with extreme soaring capacities. *Journal of Animal Ecology* 89, 20–28. doi:10.1111/1365-2656.13121 (2020).

(3) Fuster, J. M. Frontal lobe and cognitive development. *Journal of neurocytology* 31, 373–385.

doi:10.1023/A:1024190429920 (2002).

(4) Ramsaran, A. I., Schlichting, M. L. & Frankland, P. W. The ontogeny of memory persistence and specificity. *Developmental Cognitive Neuroscience* 36, 100591. doi:10.1016/j.dcn.2018.09.002 (2019).

(5) Williams, H. J., Duriez, O., Holton, M. D., Dell’Omo, G., Wilson, R. P. & Shepard, E. L. C. Vultures respond to challenges of near-ground thermal soaring by varying bank angle. *Journal of Experimental Biology* 221, jeb174995. doi:10.1242/jeb.174995 (Dec. 2018).

(6) Williams, H. J., King, A. J., Duriez, O., Börger, L. & Shepard, E. L. C. Social eavesdropping allows for a more risky gliding strategy by thermal-soaring birds. *Journal of The Royal Society Interface* 15, 20180578. doi:10.1098/rsif.2018.0578 (2018).

(7) Harel, R., Horvitz, N. & Nathan, R. Adult vultures outperform juveniles in challenging thermal soaring conditions. *Scientific reports* 6, 27865. doi:10.1038/srep27865 (2016).

(8) Ruaux, G., Lumineau, S. & de Margerie, E. The development of flight behaviours in birds. Proceedings of the Royal Society B: Biological Sciences 287, 20200668. doi:10.1098/rspb.2020.0668 (2020).

(9) Bolker, B., Warnes, G. R. & Lumley, T. Package gtools. *R Package "gtools" version 3.9.4* (2022).

(10) Bortolotti, G. R. Age and sex size variation in Golden Eagles. *Journal of Field Ornithology* 55,

54–66 (1984).

(11) Katzner, T. E., Kochert, M. N., Steenhof, K., McIntyre, C. L., Craig, E. H. & Miller, T. A. *Birds of the World* (eds Rodewald, P. G. & Keeney, B. K.) chap. Golden Eagle (Aquila chrysaetos), version 2.0. doi:10.2173/bow.goleag.02 (Cornell Lab of Ornithology, Ithaca, NY, USA, 2020).

(12) Bloom, P. H. & Clark, W. S. Molt and sequence of plumages of Golden Eagles and a technique for in-hand ageing. *North American Bird Bander* 26, 2 (2001).